# Transmission delays and frequency detuning can regulate information flow between brain regions

**Aref Pariz**[1,2], **Ingo Fischer**[2], **Alireza Valizadeh**[1,3]*, **Claudio Mirasso**[2]*

**1** Department of Physics, Institute for Advanced Studies in Basic Sciences (IASBS), Zanjan, Iran, **2** Instituto de Física Interdisciplinar y Sistemas Complejos IFISC (UIB-CSIC), Campus Universitat de les Illes Balears, Palma de Mallorca, Spain, **3** School of biological sciences, Institute for research in fundamental sciences (IPM), Tehran, Iran

* valizade@iasbs.ac.ir (AV); claudio@ifisc.uib-csic.es (CM)

**Data Availability Statement:** All relevant data are within the manuscript. Codes used to simulate and to perform the analysis in this study are available from https://github.com/ITNG/ParizPLOS2021.

## Abstract

Brain networks exhibit very variable and dynamical functional connectivity and flexible configurations of information exchange despite their overall fixed structure. Brain oscillations are hypothesized to underlie time-dependent functional connectivity by periodically changing the excitability of neural populations. In this paper, we investigate the role of the connection delay and the detuning between the natural frequencies of neural populations in the transmission of signals. Based on numerical simulations and analytical arguments, we show that the amount of information transfer between two oscillating neural populations could be determined by their connection delay and the mismatch in their oscillation frequencies. Our results highlight the role of the collective phase response curve of the oscillating neural populations for the efficacy of signal transmission and the quality of the information transfer in brain networks.

## Author summary

Collective dynamics in brain networks are characterized by a coordinated activity of their constituent neurons that lead to brain oscillations. Many evidences highlight the role that brain oscillations play in signal transmission, the control of the effective communication between brain areas, and the integration of information processed by different specialized regions. Oscillations periodically modulate the excitability of neurons and determine the response of those areas receiving the signals. Based on the communication through coherence (CTC) theory, the adjustment of the phase difference between local oscillations of connected areas can specify the timing of exchanged signals and therefore, the efficacy of the communication channels. In this respect, an important factor is the delay in the transmission of signals from one region to another that affects the phase difference and timing, and consequently the impact of the signals. Despite this delay plays an essential role in CTC theory, its role has been mostly overlooked in previous studies. In this manuscript, we concentrate on the role that the connection delay and the oscillation frequency of the populations play in the signal transmission, and consequently in the effective connectivity,

**Funding:** The work of AP, IF and CM was partially supported by the Spanish State Research Agency, through the Severo Ochoa and Maria de Maeztu Program for Centers and Units of Excellence in R&D (MDM-2017-0711) and the MINECO (Spain) through project TEC2016-80063-C3 (AEI/FEDER, UE). IF and CM acknowledge support from the Ministerio de Ciencia e Innovación through projects PID2019-111537GB-C21/AEI/10.13039/501100011033 and PID2019-111537GB-C22/AEI/10.13039/501100011033, respectively. A.V. acknowledges partial support from Iranian Cognitive Sciences and Technologies Council, under the grant No. 832. The funders had no role in study design, data collection and analysis, decision to publish, or preparation of the manuscript.

**Competing interests:** The authors have declared that no competing interests exist.

between two brain areas. Through extensive numerical simulations, as well as analytical results with reduced models, we show that these parameters have two essential impacts on the effective connectivity of neural networks: First, that the populations advancing in phase to others do not necessarily play the role of the information source; and second, that the amount and direction of information transfer dependents on the oscillation frequency of the populations.

## Introduction

A typical sensory response process in the nervous system consists of the active selection of relevant inputs, the segregation of the different features of the input, and the integration of the information leading to the right action. All these stages depend on the flexibility in the information routing, as well as in an efficient communication between different regions of the nervous system. However, the circuit and the dynamical mechanism explaining the fast reconfiguration of the effective pattern of communication and the information transfer in the neural systems have so far not been satisfactorily understood.

One interesting and widespread proposal is that in the presence of neural oscillations, communication patterns can be controlled by adjusting the phase relationship between local oscillations of different brain regions [1–8]. In the brain, synaptic interactions lead to correlated activity of the neurons and the appearance of successive epochs of high and low excitability, characterized by collective neuronal oscillations in different frequency bands [9–15]. Neural oscillations establish intermittent windows of high and low excitability, giving rise to a time-dependent response of the system to the inputs from other brain regions. According to the communication through coherence (CTC) hypothesis, it is possible to adjust the phase relationship between two regions to activate and deactivate the communication channel [1, 16–18] or continuously vary the efficacy of the channel [5]. While in the original proposal, the widespread variability in neural systems and the inconsistency of the coherence across time and space were ignored, recent studies showed that the mechanism works if oscillations are not persistent and even if the locking is not stable [19, 20].

The diversity and the time-dependency of the phase relationships reported in experiments [5, 21–23] are supposed to underlie the rich variety of communication patterns in the nervous circuits. Several experimental and computational studies have shown that those regions that phase advance others act as leaders and can efficiently transmit information to the laggard regions [20, 24–26]. It has been shown that the presence of mismatch in the natural frequencies, i.e. when uncoupled, of interconnected neural populations, can give rise to a finite phase difference and a directional information transfer, that is, nodes with higher natural frequency transmit information to those with lower frequency when the connection delay is neglected [20, 26]. However, one of the key parameters which determine synchronization and the phase relationship between coupled oscillators is the interaction delay due to the finite time of transmission of signals between the oscillators. Since the synchronization and the phase relationships determine the effective routes for information transfer, and the synchronization properties depend on the interaction delays, an important question arises: How do transmission delays in brain circuits affect the effective communication patterns?

In many theoretical and computational studies in networks of coupled dynamical systems, delays are disregarded mainly due to the analytical complexity and computational burden. However, delays might have a crucial impact on the collective properties of distributed dynamical systems [6, 7, 27–31]. In the brain, delays in the transmission of signals between neurons

and neural populations are quite heterogeneous and cover a wide range of values, from milliseconds to tens of milliseconds [32, 33]. So they can be of the same order or larger than some important neural time scales, for example, the integration time of the membrane, the period of gamma oscillations, or even of other bands, and temporal window for spike timing dependent plasticity [13, 34–40]; therefore cannot be ignored. The role that the delay play in the dynamics and phase-locking in large scale brain networks has been studied in recent years [41]. By using low-dimensional models, like neural mass or phase oscillator models, the regions of stability for in-phase, anti-phase, and out-of-phase-locking have been obtained for homogeneous networks and in the presence of heterogeneity in the connectivity of the network and in the natural frequency of the nodes [42]. Of importance, is the phase-locking that occurs in a phase different from 0 or $\pi$, since it results in a preferred direction of effective connectivity. This directional effective connectivity could be induced by imposing a mismatch between the natural oscillating frequencies of the nodes [33] and/or through a symmetry breaking mechanism in homogeneous circuits [24].

In this manuscript, we study the conditions for an effective communication between two coupled neural populations by systemically varying the interaction delay and mismatch of their natural oscillation frequencies. Our results show that for small delays, the information encoded in the population with higher natural frequency is transmitted to the other population while when the information is encoded in the population oscillating at a lower frequency, the other population is unable to receive the information, in agreement with previous results [26]. We find, however, that this is not always the case and the degree and the direction of the effective communication between populations depend, in general, on the interaction delay. Moreover, in the presence of frequency mismatch, symmetric information transmission, and efficient transmission in the reverse direction (from slow to fast) are also possible for certain range of delays.

Using a formulation based on coupled phase oscillators and the phase response curve we were able to provide a general framework to predict how the pattern for effective communication between two coupled oscillators changes with the delay and frequency mismatch. These novel findings provide a theoretical basis to understand how the information is transmitted in brain circuits along different channels and directions and over different frequency bands.

## Materials and methods

### Neuron model

In our simulation we used the Hodgkin-Huxley (HH) neuron model [43]. The evolution of the membrane potential and gate variables are given by:

$$
\begin{aligned}
C\frac{dv}{dt} &= I_{ext} + I_{syn} - g_K n^4 (v - E_K) \\
&\quad - g_{Na} m^3 h(v - E_{Na}) - g_L(v - E_L) \\
\frac{dn}{dt} &= \alpha_n(v)(1 - n) - \beta_n(v) \\
\frac{dm}{dt} &= \alpha_m(v)(1 - m) - \beta_m(v)m \\
\frac{dh}{dt} &= \alpha_h(v)(1 - h) - \beta_h(v)h
\end{aligned}
\tag{1}
$$

where $I_{ext}$ and $I_{syn}$ are the input and synaptic currents, respectively. The $\alpha_x$ and $\beta_x$, $x \in (n, m$

and $h$) are defined as below

$$
\begin{aligned}
\alpha_n(v) &= \frac{0.01(v+55)}{1-exp(-0.1(v+55))} \\
\beta_n(v) &= 0.125exp(-0.0125(v+65)) \\
\alpha_m(v) &= \frac{0.1(v+40)}{1-exp(-0.1(v+40))} \\
\beta_m(v) &= 4exp(-0.0556(v+65)) \\
\alpha_h(v) &= 0.07exp(-0.05(v+65)) \\
\beta_h(v) &= \frac{1}{1+exp(-0.1(v+35))}
\end{aligned}
\tag{2}
$$

The values of the parameters are given in Table 1.

The synaptic current of the i-th neuron ($I_{syn}^i$) is given by:

$$
\begin{aligned}
I_{syn}^i(t) &= \sum_j g_{ij} S_{ij}(t)(v_i - E_{syn}^j) \\
S_{ij}(t) &= \frac{1}{A}\left(exp(-(t-t_j^*-t_d^{ij})/\tau_r)\right. \\
&\quad \left. - exp(-(t-t_j^*-t_d^{ij})/\tau_d)\right) \\
A &= \left(\frac{\tau_r}{\tau_d}\right)^{\frac{\tau_r}{\tau_d-\tau_r}} - \left(\frac{\tau_r}{\tau_d}\right)^{\frac{\tau_d}{\tau_d-\tau_r}}
\end{aligned}
\tag{3}
$$

$v_i$ is the membrane potential of the post-synaptic neuron and $E_{syn}^j$ is its reversal synaptic potential. $S_{ij}$ is a double-exponential function, modeling the efficacy of the chemical synapses mediated by $AMPA$ and $GABA_a$ receptors. $t_j^*$ is the time at which the pre-synaptic neuron spikes

**Table 1. Simulation parameters.**

| Parameter | Value | Description |
|---|---|---|
| C | $1\ \mu F/cm^2$ | Capacitance |
| $g_K$ | $36\ mS/cm^2$ | K conductance |
| $g_{Na}$ | $120\ mS/cm^2$ | Na conductance |
| $g_L$ | $0.3\ mS/cm^2$ | Leak conductance |
| $E_K$ | $-77\ mV$ | K reversal potential |
| $E_{Na}$ | $50\ mV$ | Na reversal potential |
| $E_L$ | $-54.4\ mV$ | Leakage reversal potential |
| $E_{syn}^E$ | $0\ mV$ | Excitatory reversal potential |
| $E_{syn}^I$ | $-80\ mV$ | Inhibitory reversal potential |
| $t_d^{inner}$ | $0-14\ ms$ | Inner population delay of excitatory unit |
| $t_d^{intra}$ | $0.5\ ms$ | Intra population axonal delay |
| $\tau_d$ | $3\ ms$ | Synaptic decay time |
| $\tau_r$ | $0.5\ ms$ | Synaptic rise time |
| $I_{ext}$ | $10-12\ \mu A/cm^2$ | Injected current |
| $\bar{\mu}, \sigma$ | $0, 0.5\ \mu A/cm^2$ | Mean and variance of Gaussian white noise |
| $g_{EE}$ | $3.75\mu S/cm^2$ | Synaptic weight, E → E |
| $g_{EI}$ | $15\mu S/cm^2$ | Synaptic weight, I → E |
| $g_{IE}$ | $7.5\mu S/cm^2$ | Synaptic weight E → I |
| $g_{II}$ | $15\mu S/cm^2$ | Synaptic weight I → I |

and $t_d$ is the axonal delay between pre- and post-synaptic neurons. The synapses' parameters and synaptic weights ($g_{ij}$) are given in Table 1. We numerically solved the equations using the Milshtein algorithm [44] with an integration step $dt$ = 0.01 ms.

## Population architecture

Each of our populations was composed of 100 neurons with 80% excitatory and 20% inhibitory neurons. The connectivity within the population was chosen random and with probability of 10%. The connectivity between populations was random (but just among excitatory neurons) with probability 5%. The intra population delay was taken 0.5 $ms$ while that of between populations was varied from 0 to 14 $ms$.

## Input signals

We injected a constant current (varied from 10 to 12 $\mu A/cm^2$) and an uncorrelated Gaussian white noise ($\bar{\mu} = 0 \ \mu A/cm^2$, $\sigma = 0.5 \ \mu A/cm^2$) to each neuron ($I_{ext}$ in Eq (1)). Depending on the value of the input current, neurons fired within a frequency range of 70-73 Hz.

To test the quality of the signal transmission we first injected slow (5 Hz) non-periodic signals (see Theoretical background subsection) into the excitatory neurons of the host (or sender) population. By running simulations in the absence of an external signal, we recorded the peaks of the network activity and the mean interval between the successive peaks was taken as the period of oscillation (note that by calculating the oscillation frequency in this way, its value might be different from the average frequency of firing of the constituent neurons [45]). In the fast-signal case, we applied to all the excitatory neurons a single pulse at a certain phase of the period. The phase of the impact of the pulse changed in each numerical realization by dividing the oscillation cycle into 50 segments. In each realization, we applied a rectangular pulse of amplitude $I_{pulse}$ = 0.25$\mu A/cm^2$ and width of 2 (ms).

## Analysis

In our analysis, we calculated the firing rate (multi-unit activity (MUA)) of the populations by using a Gaussian time window with standard deviation $\sigma$ = 2 $ms$ and $\sigma$ = 100 $ms$ for the fast and the slow modulation, respectively. By sliding the Gaussian time window, we calculated the weighted sum of the number of spikes in the window and taken the value of it in the related time as the instantaneous firing rate at the center of the time window.

**Cross-covariance.** The cross-covariance quantifies the similarity between two vectors. We used the un-biased and un-normalized value of the cross-covariance at zero lag (ZLC), to quantify the similarity between the firing rates of the receiver population with the input signal that we injected on the excitatory neurons of the sender (host) population. We assumed that if the signal was transmitted to the second population, its firing rate should follow the signal. In the figures, we plotted the un-biased and un-normalized value of the cross-covariance at zero lag (ZLC).

**Coherency.** We took the normalized average of the amplitude of the network activity as the coherency factor. The network activity of the populations was calculated using a Gaussian time window with standard deviation $\sigma$ = 2($ms$). 20 successive peaks of the populations firing rate were averaged and normalized by the maximum value of the amplitude (when all the neurons fired at the same time) to calculate a normalized coherency index $C$.

**Delayed mutual information.** To characterize the signal transmission and to define the effective connectivity between the coupled populations, we calculated the time delayed mutual information (dMI) [46]. The dMI quantifies the causal relationship between the activities of the coupled populations. The time delayed mutual information is computed based on the

Shannon's entropy for two vectors as:

$$dMI_{ij}(d) = \delta MI(X_i(t), X_j(t+d)))$$
$$= H(X_i(t)) + H((X_j(t+d)))$$
$$- H(X_i(t), X_j(t+d))$$

(4)

where $H(X_i) = -\sum_{k \in X_i} P_k log(P_k)$ is the marginal entropy of $X_i$ and log is base 2 logarithm, and $P_k$ is the probability of occurrence of event k. The Joint entropy in Eq (4) is calculated as

$$H(X_i(t), X_j(t+d)) = -\sum_{n \in X_i(t)} \sum_{m \in X_j(t+d)} P_{n,m} log(P_{n,m}).$$

(5)

In our case, the vectors $X_i$ and $X_j$ represented the network activity (firing rate) of the two populations. By integrating the dMI for positive lags, we quantified the amount of information that was transmitted from *i* to *j*; integrating over negative lags, we computed the transferred information in the opposite direction. Subtracting these two values, we obtained the net information transferred between the two populations.

**Phase response curve (PRC) of a population.**   To find the response of a population to an injected pulse, we proceeded in a similar way as it is done for calculating the PRC of a single neuron. We partitioned the time between two successive peaks of network activity into 30 segments. Keeping all other parameters unchanged, we applied a rectangular pulse at a specific phase (segment) on all the excitatory neurons of the population. We defined the PRC as the difference between the instantaneous oscillation period of the population without and with the injected pulse, multiplied by a factor $2\pi/T$ (see Fig 1). The width and amplitude of the rectangular pulse that we used were 2 *ms* and 1 $\mu A/cm^2$, respectively.

## Results

We start by numerically investigating the role of the combination of transmission delay and frequency mismatch on the effective communication between two bidirectionally connected neural populations (see the Materials and methods section for details). Each population consists of N neurons (80% excitatory and 20% inhibitory) modeled by the Hodgkin-Huxley (HH) equations. The intra-population connectivity probability is 10% for all types of connections. Long-range excitatory projection connect the excitatory neurons of the two populations with a probability of 5%. The synaptic currents were assumed to be mediated by *AMPA* and $GABA_A$ receptors, were modeled by a double-exponential function with the synaptic rise and decay time equal to 0.5 ms and 3 ms, respectively, for both type of synapses. The delay between any pair of connected neurons inside the populations is assumed to be 0.5 *ms*. All neurons received an injected constant current and a Gaussian white noise with mean $\mu$ and variance $\sigma$.

In Fig 1A and 1B, the coherency and the oscillation frequency of an isolated population are shown, when changing the mean and the variance of the external noise. Raster plots of the spiking activity of the neurons, and the population activity of the network are shown in Fig 1C, for the two values of parameters depicted by dots in Fig 1A and 1B. In the rest of the manuscript, we fixed the parameters to the values used in Fig 1C, bottom panel, which lead to a coherency value $C \simeq 0.8$ and frequency $f \simeq 70$ Hz.

### Phase-locking between populations

We now concentrate on the case of two populations connected with a given delay time (Fig 2). The mean input current into population 2 is fixed at $11\mu A/cm^2$, while the mean input current into population 1 is $(11 \pm \Delta) \mu A/cm^2$, where $\Delta$ introduces a frequency mismatch (detuning)

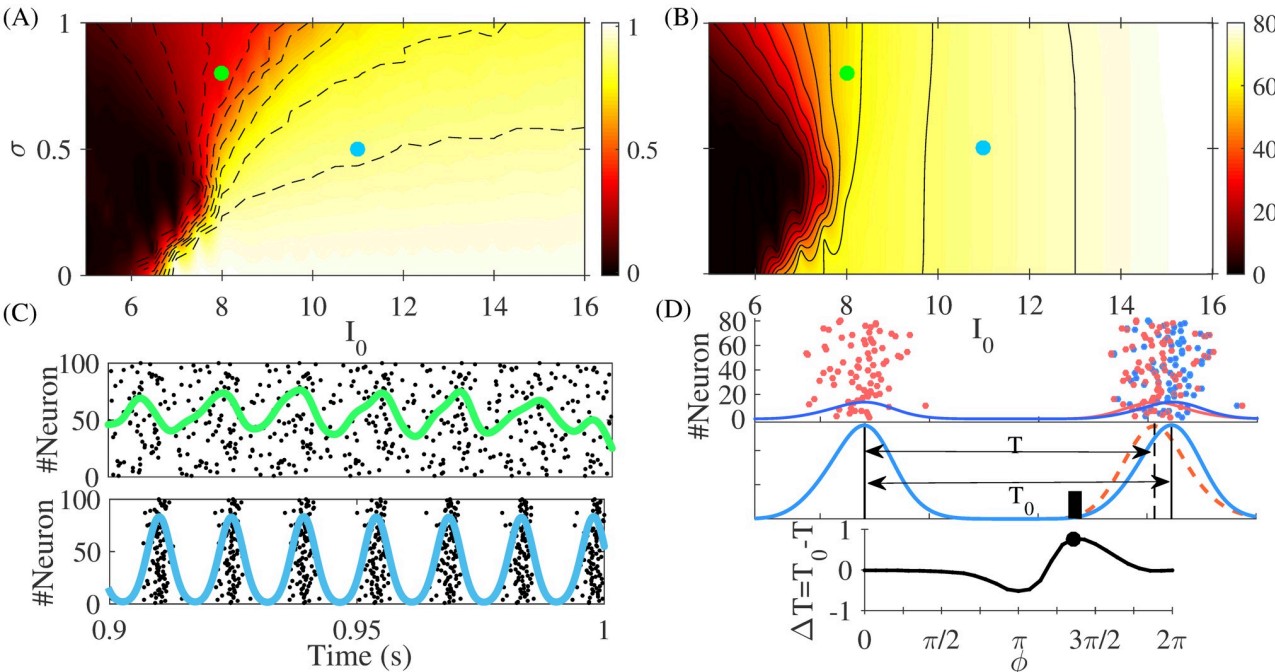

**Fig 1. Properties of a single population.** The coherency (panel A) and the oscillation frequency (panel B) of the population activity is plotted in the mean input current ($I_0$) versus noise amplitude ($\sigma$) plane. In panel C the raster plots and the activity of a single population for the two points, depicted by green and blue dots, in panels A and B. In the rest of the simulations we used the set of parameters considered in lower plot of panel C with the coherency value $C = 0.80$. In panel D we have schematically shown how the PRC of a population (pPRC) is calculated. The interval between two successive peaks of the network activity before ($T_0$) and after the injection of the pulse ($T$) is recorded. The difference between these two intervals is defined as the pPRC after multiplication by $2\pi/T$, as is explained in the text.

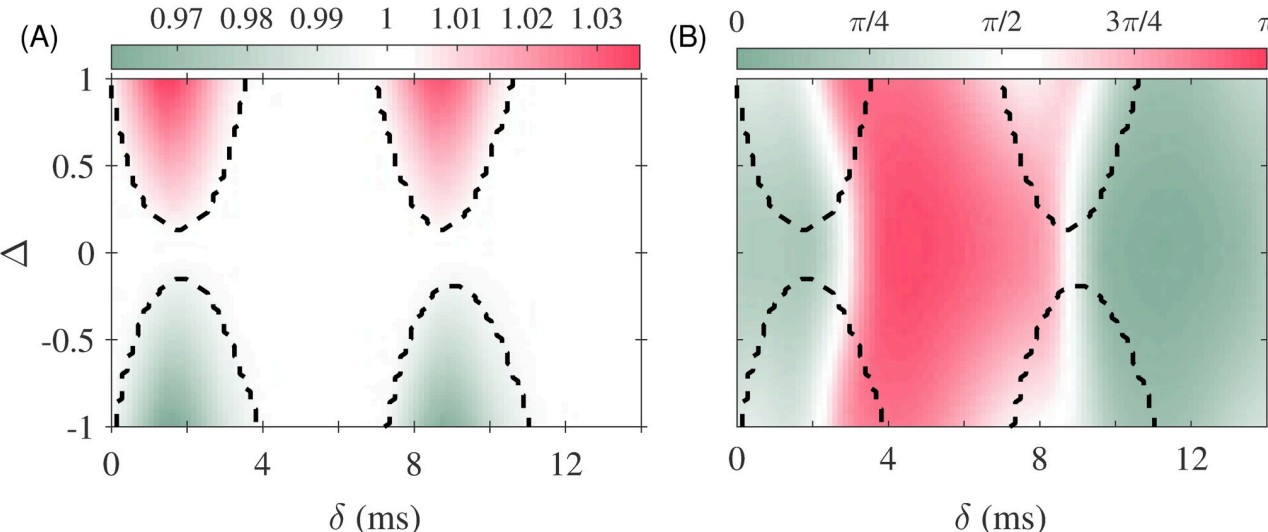

**Fig 2. Phase-locking of the populations.** (A) The ratio of the oscillation frequencies of the two connected populations is shown for different values of the input current mismatch ($\Delta I$) and delay ($\delta$). The locking zone can be distinguished by the white color whose borders are shown by dashed lines. In (B) the phase difference of the two populations is shown. Note that phase difference is only well-defined within the locking zones.

between the two populations. We hypothesized that the theory of coupled oscillators can qualitatively predict the properties of a system of two populations connected via long-range projections [6, 27, 31, 43, 47–49]. Then the synchronization between oscillations of the two populations would be determined by the coupling strength and transmission delay of the connections as well as the mismatch between the natural oscillation frequencies of the populations [20, 30]. In this study, we fixed the strength of the long-range connections and varied the frequency mismatch and the time delay. We observed that the locking window, determined by the frequency mismatch for which the system remains in the phase-locked regime, depended on the delay, as expected (Fig 2A). Within the locking zone, the two coupled populations oscillated at the same frequency, while the phase difference between their oscillations changed with the delay and the frequency mismatch (Fig 2B). Such a varying phase difference affects the signal and information transmission between the two populations as is shown below.

## Transmission of slow (rate) signals

To evaluate the ability of the system to transmit information, we applied a time-dependent signal on one of the populations (the sender) and check how this signal was transmitted to the other population (the receiver). We considered the two widespread neuronal coding schemes, rate and spike-time coding [50, 51]. We did this by applying two different types of signals. In the first case, we modulated the oscillation frequency of the sender population by using a time-dependent input current (only on the excitatory neurons) whose frequency was much lower than the oscillation frequency of the populations (Fig 3B–3D).

To assess the quality of the signal transmission, we first extracted the instantaneous oscillation frequency of the two populations, and then calculated the cross-covariance between the

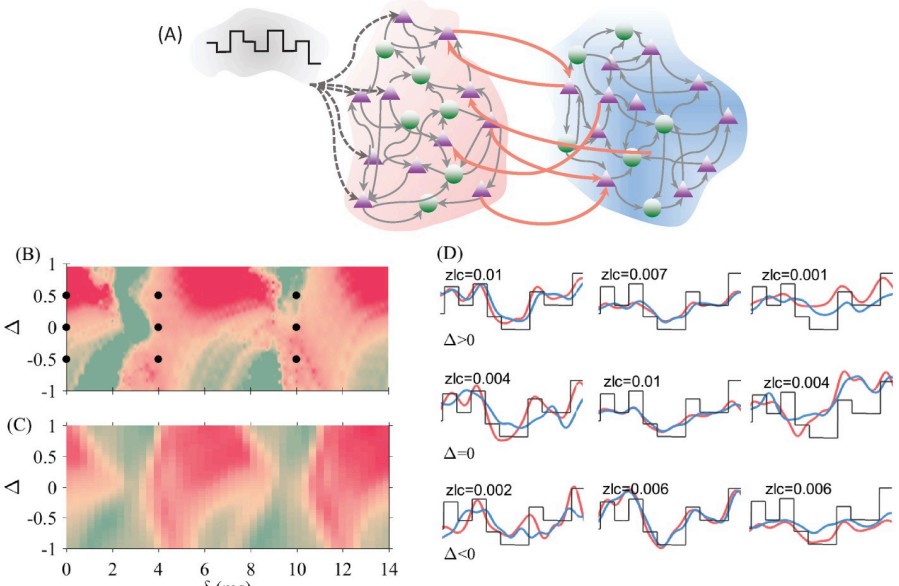

**Fig 3. Signal transmission and information transfer between two bidirectionally coupled populations.** (A) Schematic diagram of the network connectivity. Only excitatory neurons between the two populations are connected. Zero-lag cross-covariance (ZLC) between the firing rate of the second population with the input signal (B), and the delayed mutual information between firing rates of two populations (C) are shown by color code. In (D) the firing rates of the two populations and the input signal are plotted, for the parameters values marked with black dots in (B). The value of ZLC is shown in each panel. The offset and the amplitude of the external signal were varied for a better comparison.

rate of the receiver population and the signal (see Materials and methods). The result is shown in Fig 3B. The red color indicates a good transmission quality while the green color denotes that the transmission is degraded. Some aspects are to highlight in Fig 3D. A dominant red color is observed for positive detuning, indicating that the signal is better transmitted when the sender has a higher natural frequency than the receiver. This occurs for most values of the transmission delays. Likewise, areas of weak transmission can be seen for positive detuning over some specific range of delays. For negative detuning, signal transmission can also occur from the population with lower oscillation frequency to that of high oscillation frequency for certain delays with a relatively good, although not maximal, quality. There are also some delay values that permit transmission in both directions with a relatively good transmission quality. In any way, it is evident that the zero detuning case is not an optimal choice to transmit the signal. It is shown in Fig 3C that similar results are obtained when computing the delay mutual information (discussed in the Information transfer section).

Time traces of the evolution of the populations' rate superimposed on the signal, are shown in Fig 3D for different delays and frequency mismatches (black dots in Fig 3B). It is seen that for small delays the signal transmits better from the population that oscillates at a higher frequency to the one that oscillates at a lower frequency, as reported in previous studies (Fig 3B and the left column of Fig 3D) [19, 26, 52]. However, this no longer holds for larger delays. We observe that, for some values of the delays, the quality of the transmitted signal can be almost the same for both positive or negative values of the detuning (Fig 3B and middle column of Fig 3D) or can be even better for negative values of the detuning, i.e., when the sender population oscillates at a lower frequency (Fig 3B and the right column of Fig 3D).

## Transmission of pulse packets

In this case, we applied a single pulse packet on all excitatory neurons of the sender population at a certain phase (between 0 and $2\pi$ over one cycle of the oscillation) and measured the change in the response in both the sender and receiver populations. The phase at which the pulse was applied was varied to cover the whole $2\pi$ range. The effect of these pulses in the sender population was characterized by the local population phase response curve (pPRC) while that in the receiver population was quantified by the non-local phase response curve (nPRC), where the latter is a measure of the signal transmission quality.

In the different panels of Fig 4A the pPRC and the nPRC are shown for different values of delay and frequency detuning. Green curves in Fig 4A show the prediction of the analytical results based on the multiplication of the pPRC (red curve) and the absolute value of its derivative, at the time at which the spikes of the sender populations arrive to the receiver population (Eq (10)). It can be seen that for certain delay values, the nPRC has a finite amplitude which indicates that the signal is transmitted while for other delays the nPRC is flat indicating that the signal is not detected by the receiver population.

The results of the simulation of spiking neurons is also shown in Fig 4A (blue dots). A qualitative agreement is observed between the theoretical and numerical results for most values of the frequency mismatch and delay. For the comparison, we have shown the maximum values of the two curves in Fig 4B and 4C. Although a good agreement is seen for most values of the delay and mismatch, the results do not perfectly match. We hypothesize that the main reason for the difference in some cases is that, the spiking activity of the coupled populations are not perfectly synchronous while the PRC approximation works well for pulsatile interaction between the two populations. The temporal width of the population activity depends on the coherency of the populations, and could also be affected by the external signal. A finite

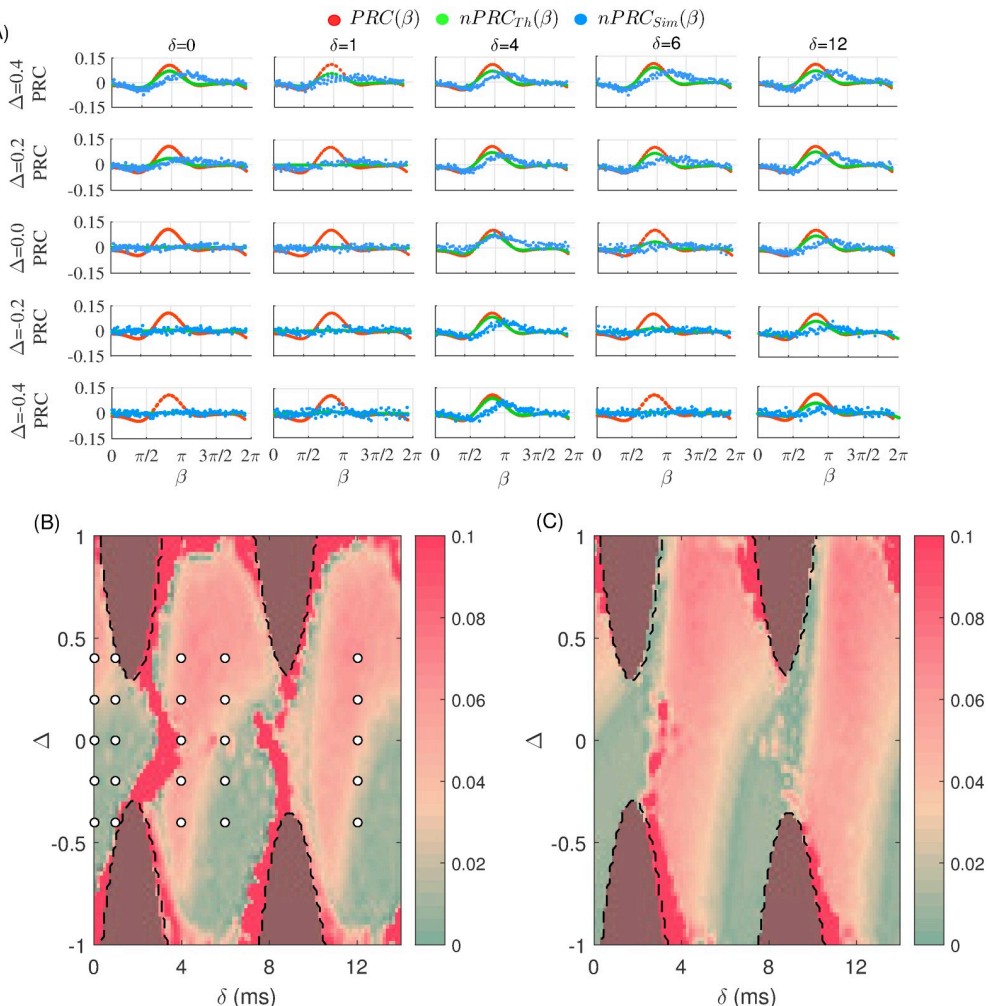

**Fig 4. Response of the network to fast signals.** (A) The PRC of the sender population (pPRC; red) and the non-local PRC (nPRC) resulted from simulation (sky-blue) and theory (green) is shown for different values of the delay and detuning. The horizontal axis shows the phase $\beta$ at which the pulse impacts the sender population. The green curve shows the analytical prediction for the response calculated as the value of the derivative of PRC, multiplied by the value of PRC (Eq(10)). In panels (B) and (C) we plot the maximum values of nPRC in the delay and detuning phase plane as predicted by theory (green curve in (A)) with those obtained through simulation (blue dots in (A)), respectively. The results are only shown for phase-locked regions (within the dashed lines) since the analytical results are not valid beyond these regions.

temporal width of the spiking activity of the sender population could make a deviation from the analytical prediction in this case.

It can also be seen that the results qualitatively agree with those obtained for slowly varying signals (rate modulation; see Fig 3), i.e., for small delays, the signals are more efficiently transmitted from the fast to the slow oscillating population. For larger delay values instead, symmetric transmission or even a better transmission in the reverse direction are found. It should be noted that, as occurs in the case of slow modulation, better signal transmission is found in general for positive values of the detuning (higher oscillation frequency of the sender population) as compared to the case of negative values of the detuning (lower oscillation frequency of the sender population) when changing the connection delay.

## The PRC qualitatively predicts information transmission flow

Our previous results highlight the importance of the PRC analysis. Indeed, the response of the receiver population to a perturbation applied to the sender population depends on the excitability state of the both populations at the time they receive the perturbation. The value of the PRC at the phase at which the external pulse impacts the sender populations determines the local response and the effect of the pulse on the sender population. This effect is quantified by the change of timing of the next spiking activity of the sender population, therefore, if the receiver population receives the perturbation (after a transmission time $\delta$) in a phase at which the time-derivative of the PRC is non-negligible (see blue solid curve in Fig 5A), then the receiver population detects the perturbation. Otherwise, the receiver population does not detect the change in the timing of the incoming spiking activity and the perturbation is filtered out. In Fig 5A we have schematically shown the situation to demonstrate how the transmission of the pulse packets can be quantified by analyzing the PRC of the populations. The magnitude of the time derivative of the PRC of the receiver population at the time the impact of spiking activity of the sender population quantifies the transmission.

In Fig 5B we have plotted the absolute value of derivative of the PRC of the receiver population at the time it receives the spikes from the sender population (without any external perturbation) when changing the detuning and the delay (see also the section Theoretical background). It can be seen that the results agree well with those obtained by analyzing the pulse transmission in Fig 4B, highlighting the validity of PRC analysis. Once the phase response curves of the populations are known, in the delay and natural frequency mismatch phase plane, the phase difference in the locked state can be derived and the non-local PRC can be calculated. The latter determines the quality of the signal transmission and therefore of the information in the network. In Fig 5C we also show the obtained delayed mutual information (see next section), which quantifies the magnitude of information transfer from the sender to

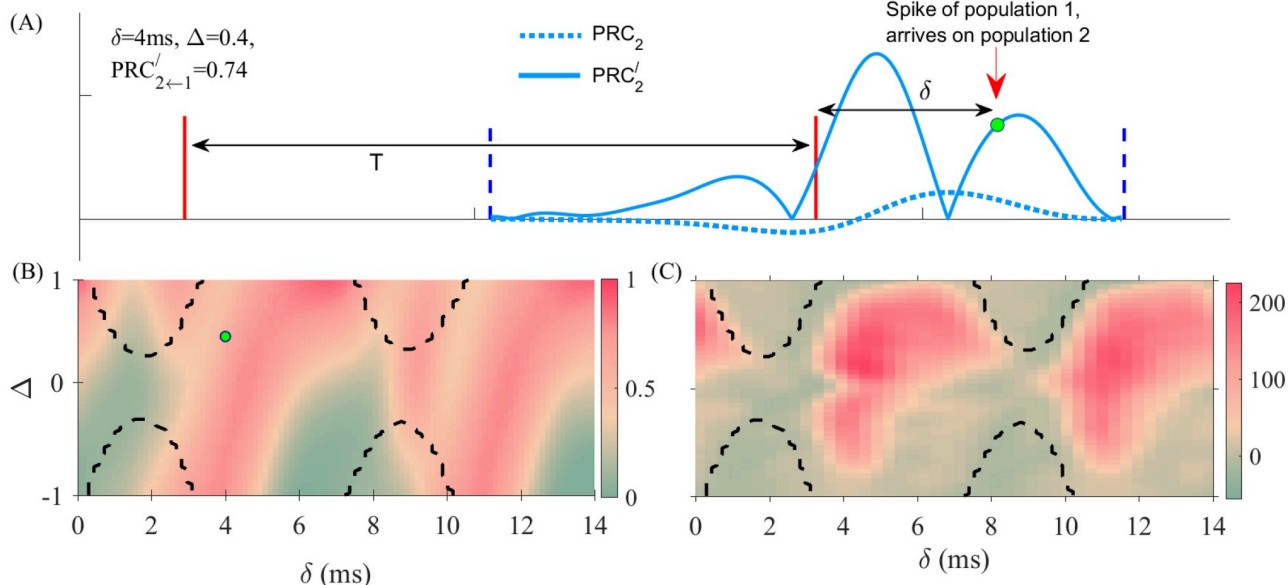

**Fig 5.** (A) Schematic plot of the PRC (dotted lines) and its time derivative (solid lines) represented between two consecutive oscillation peaks of the populations for $\delta = 4$ and the detuning value $\Delta = 0.4$. The $PRC'_{2\leftarrow 1}$, is the absolute value of $PRC'$ of the population 2 (solid blue line) when it receives the spikes of population 1. This measure predicts the quality of the transmission. (B) The absolute value of derivative of the PRC of the receiver population at the time it receives the spikes from the sender population, is plotted in the delay-detuning phase plane. (C) The $\delta MI$ between the firing rates of the populations computed in the absence of an external signal.

the receiver population. The results show that the preferred direction of the signal and information transmission can be qualitatively predicted by the theoretical analysis of coupled phase oscillators once the PRC of the nodes is known.

## Information transfer

In the previous section, we used cross-covariance and phase response curves as measures of the quality of the transmission of external signals in the system. Since these measures are linear it is not readily clear if they can predict (and be inferred from) the causal relationship between the populations and the direction of the information flow between them in the presence and in the absence of the external signal. To clarify this point, we compute the delayed mutual information (see Materials and methods, Eq(4)) between the two populations. This measure quantifies the information flow regardless of how the information is encoded and decoded [46].

Previous studies have shown that, in the absence of transmission delay, a frequency mismatch between the oscillations of the two populations breaks the symmetry of the information flow favoring the fast-to-slow direction [19, 26, 52]. As it can be seen in Figs 3D and 5C (in the presence and in the absence of the external signal, respectively) the direction of the information flow changes with the delay and frequency mismatch in a qualitatively similar manner as for the signal transmission, indicating that the quality of the signal transmission can accurately predict the direction of information flow and vice versa.

It is worth mentioning that similar results can be obtained when the networks oscillate in another frequency, being the only difference that the range of delays scales with the carrier frequency. This is a remarkable result due to its functional importance: When taking into account the transmission delays, the quality of signal transmission depends on the (carrier) frequency.

## Effect of asymmetric connectivity

The connections between brain regions are mostly asymmetric [53, 54]. It was previously shown that in heterogeneous networks when the connections are chosen from a long-tale distribution, the nodes with stronger connections lag behind the weaker connected nodes [33]. It means that the phase differences can also be affected by the asymmetry in synaptic strengths, besides the natural frequency mismatch and communication delay which was explored above. Therefore, here we inspect how the change in the phase differences due to the asymmetric connections affects the signal and information transfer between neuronal populations.

We first explored the results for a feedforward network (no feedback connection) from the receiver to the sender. The computed mutual information transfer reveals that the transmission is independent of the delay, as expected, and is determined by the mismatch between the oscillation frequencies of the two populations (Fig 6A). For all delay values, the positive mismatch (higher frequency of the sender population) yields a better information transfer.

We then fix the detuning at a positive value and the connection from sender and receiver to $g_{for}$, and vary the strength of the connection from the receiver to the sender from zero to $2g_{for}$. The effect of the delay becomes more evident for an increasing feedback strength (Fig 6B). For $g_{back} < 0.5g_{for}$ the information transmission remains from the sender to the receiver almost independently of the delay. For $g_{back} > 0.5g_{for}$ we found windows of delays where the information transfer from the sender to the receiver is considerably degraded, while in other ranges the transmission is facilitated. Interestingly, the presence of the feedback connections facilitate the transmission for some ranges of delay and degrades the transmission for some other ranges.

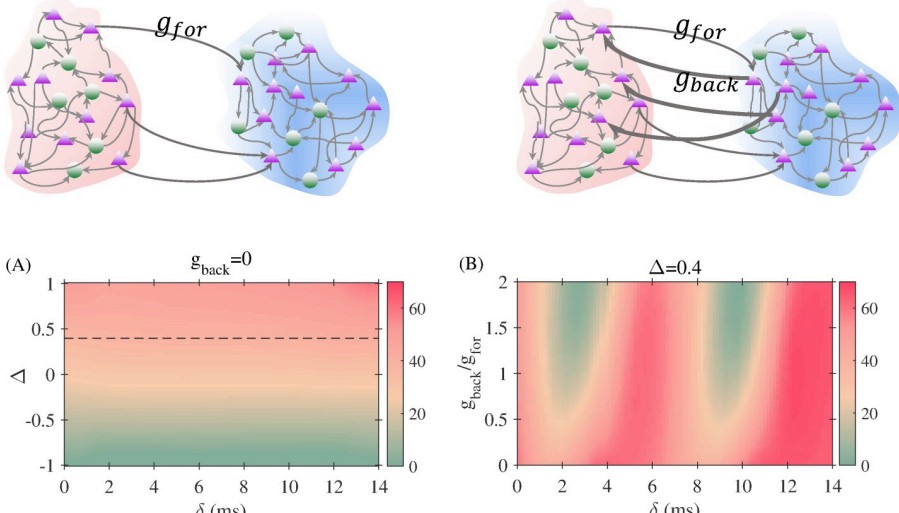

**Fig 6. Effect of connection asymmetry on the information transfer.** In (A) the connection is unidirectional from the sender to the receiver. The mutual information transfer depends on the detuning but, as expected, is independent of the delay. In (B) the delayed mutual information is plotted, in color code, in the ratio between feedforward and feedback strength versus delay plane for a positive inhomogeneity value ($\Delta = 0.4$).

## Theoretical background

To gain insight into the mechanisms that regulate the information flow between to delay-coupled oscillating neuronal populations, we analyzed a minimal model of two coupled phase oscillators. These oscillators are characterized by their natural frequency $\omega_i$ and their phase response function $Q_i$. The evolution of the system was described by:

$$
\begin{aligned}
\dot{\theta}_1 &= \omega_1 + K_{12}Q_1(\theta_2 - \theta_1 - \delta), \\
\dot{\theta}_2 &= \omega_2 + K_{21}Q_2(\theta_1 - \theta_2 - \delta),
\end{aligned}
\tag{6}
$$

where $\theta_1$ and $\theta_2$ are the phase of the oscillators, $K_{12}$ and $K_{21}$ are the coupling strengths, and $\delta$ represents the interaction phase (that relates to the delay $\tau$ as $\delta = \omega_{locked} {}^* \tau$). This approximation is valid when delays are smaller of, or comparable with, the period of the oscillations [55]. We assumed that the strength of the connections were equal $K_{12} = K_{21} = K$ and that the response functions were the same $Q_1 = Q_2 = Q$. We then defined the new variables $\phi = \theta_1 - \theta_2$ and $\Theta = \theta_1 + \theta_2$ and found

$$
\begin{aligned}
\dot{\Theta} &= W + K\,Q(-\phi - \delta) + K\,Q(\phi - \delta) \\
&= W + K\Lambda(\phi, \delta),
\end{aligned}
\tag{7}
$$

$$
\begin{aligned}
\dot{\phi} &= \Delta + K\,Q(-\phi - \delta) - K\,Q(\phi - \delta) \\
&= \Delta + K\Gamma(\phi, \delta),
\end{aligned}
\tag{8}
$$

where, $\Omega = \omega_1 + \omega_2$ and $\Delta = \omega_1 - \omega_2$. Phase-locking is then determined by $\dot{\phi} = 0$. The phase difference in the locked state is implicitly given by

$$
\Gamma(\phi^*, \delta) = -\frac{\Delta}{K}.
\tag{9}
$$

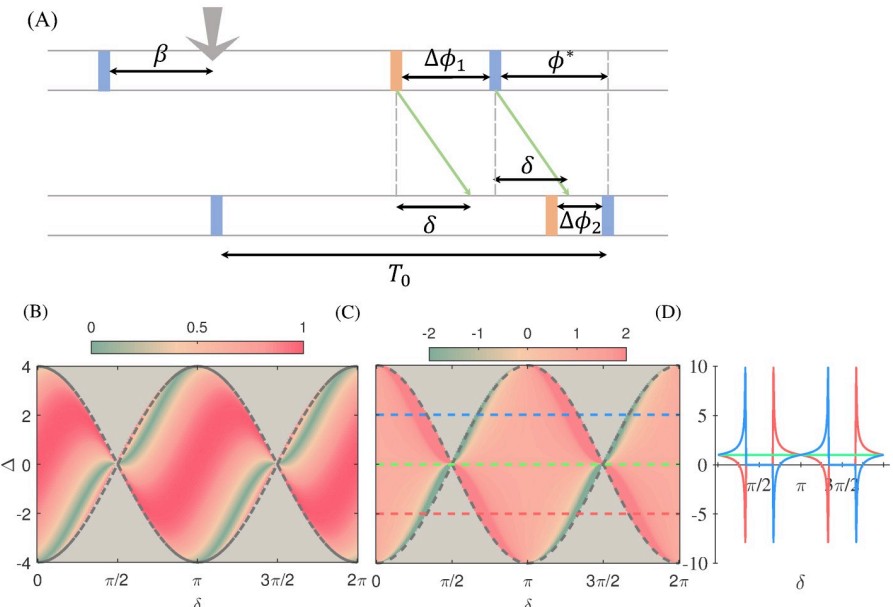

**Fig 7. Analytical results.** (A) The schematic of how a pulsatile perturbation in the first oscillator affects the phase of the second oscillator, as discussed in the text. (B) The absolute value of derivative of the PRC ($Q$), at phase ($\phi^* - \delta$). To plot this figure, we first find the locked phase difference for each value of $\Delta$ and $\delta$ in the locked regime using the Eq (9) and then we find the absolute value of $\sin(\phi^* - \delta)$. Locking zone borders is shown by dashed lines. (C) The response function $S_i$ (Eq (12)) to the input of slow signals for the case of two delay-coupled phase oscillators is plotted in color code, in the detuning and delay phase plane. In (D) the same results as (C) are shown for $\Delta = -2, 0, 2$, depicted by red, green and blue colors in (C), for the sake of clarification.

A solution of this system exists while $min(\Gamma) < -\frac{\Delta}{K} < max(\Gamma)$, and the stability condition for a locked state is given by $\frac{d\Gamma}{d\phi}\big|_{\phi^*} \leq 0$.

The main objective was to explore how a *local* external signal imposed on one of the oscillators affected the other, i.e., how the signal *transmits*. The signal appeared as a weak time-dependent perturbation on the intrinsic frequency of one of the oscillators–the sender oscillator. We then addressed the question of how the signal injected into one oscillator (the sender) affected the other oscillator (the receiver). Here we considered both tonic signals, which varied in a long time scale compared to the period of the oscillations, and pulsatile signals, that modeled the synaptic inputs whose time constants (mainly decay time) were short compared to the oscillation period of the populations. These two types of signals can be related to the rate and spike-time coding schemes in neuroscience [50, 51].

To quantify the transmission of synchronous signals, we defined a non-local phase response curve (nPRC) which was specified as the change in the phase of the receiver oscillator upon the incidence of the pulse injected into the sender one [56]. We assumed that the unperturbed oscillators were locked at the phase difference $\phi^*(\Delta, \delta)$ determined by Eq (9). The impact of a pulse at a given phase $\beta$ on the sender oscillator, changes its phase as $\Delta\phi_1 = Q(\beta)$ (see Fig 7A). This gives rise to an instantaneous change in the argument of the coupling function in the second equation of Eq (6) by $Q(\beta)$ and changes the right hand side of that equation by $Q(\beta)Q'(\phi^* - \delta)$ (where $Q'$ is derivative of $Q$ with respect to its argument), given $Q(\beta)$ is small (Fig 7A). As a result, the phase changes in the receiver oscillator (and in the nPRC) is

$$Q_{21}(\beta, \delta) = Q(\beta)Q\prime(\phi^* - \delta). \tag{10}$$

Note that in the above equation, $Q(\beta)$ quantifies how much the sender is affected by the signal,

and $Q'(\phi^* - \delta)$ quantifies to what extent the change in the phase of the sender is sensed by the receiver. In other words, the net transmission can be solely quantified by the absolute value of $Q'(\phi^* - \delta)$. As noted above, the phase difference $\phi^*$ can be calculated for any value of the delay and frequency mismatch for $Q(\theta) = \sin(\theta)$. This type of PRC serves as a canonical form for type-II excitable systems and resembles the interaction function in the Kuramoto model [57]. Fig 7B shows a generic form of the dependence of the transmission with respect to the delay and mismatch which can be sketched for any oscillator once the PRC of the population $Q$ is known. The advantage of a positive mismatch (higher intrinsic frequency of the sender) for an efficient transmission is seen for small delays $\delta < \pi/2$ while the reverse preferred direction can be seen in the range $\pi/2 < \delta < \pi$. This pattern of transmission is repeated for $\pi < \delta < 2\pi$ due to the symmetry of the PRC (this symmetry is not the case for a general oscillator). Note also that our analytical results are valid in the locked state whose domain is shown by the dashed lines. We found similar results for the case of two coupled HH oscillators/populations, although the pattern is not as symmetric as for the sinusoidal PRC.

In the second case we considered a slowly varying signal and quantified the transmission of the signal by calculating a non-local response function defined as the derivative of the rate of the collective phase change $\dot{\Theta}$ with respect to the free-running frequency of the sender oscillator $i$

$$
\begin{aligned}
S_i = \frac{d\dot{\Theta}}{d\omega_i} &= \frac{dW}{d\omega_i} + K\frac{d\Lambda}{d\omega_i} \\
&= \frac{dW}{d\omega_i} + K\frac{d\Lambda}{d\phi*} \times \frac{d\phi*}{d\omega_i}.
\end{aligned}
\tag{11}
$$

Note that the signal was assumed to be weak enough so that the system remains in the locked state, therefore, the dynamics of the collective phase is also representative of the dynamics of the receiver oscillator. The response function can be considered as a measure of the impact of the signal on the dynamics of the receiver oscillator. We will show through numerical simulation of Eq (6) that this quantity can indeed qualitatively predict the correlation between the signal and the rate of change in the phase of the receiver oscillator.

As an example we considered again $Q(\theta) = \sin(\theta)$. In this case, the phase difference $\phi^*$ in the locked state is determined by $\sin(\phi^*) = \Delta/2K\cos(\delta)$ (Eq (9)), provided that $|W/2K\cos(\delta)| \leq 1$. The non-local PRC, which quantifies the transmission of pulse signals, is proportional to $\cos(\phi^* - \delta)$. For the slow (rate) signals the response function is given by:

$$
S_i = 1 + \Delta\frac{tan\delta}{\sqrt{4K^2cos^2\delta - \Delta^2}}.
\tag{12}
$$

We also defined an imbalance measure, as the difference of response functions in two directions which quantifies the asymmetry in the signal transmission, from the high to low frequency oscillator and in the reverse direction. For the above example the imbalance was calculated as:

$$
\Delta S = 2\Delta\frac{tan\delta}{\sqrt{4K^2cos^2\delta - \Delta^2}}.
\tag{13}
$$

The analytical results for the response function $S_i$, for different values of frequency mismatch $\Delta$ and delay $\delta$, are plotted in Fig 7C and 7D. The results show that similar to the case of pulsatile signal in the locked state, for relatively small delays $0 < \delta < \pi/2$ the signal is better transmitted when it is injected into the high-frequency oscillator. Instead, for larger delays, phases $\pi/2 < \delta < \pi$, the transmission is facilitated in the reverse direction from the low- to the

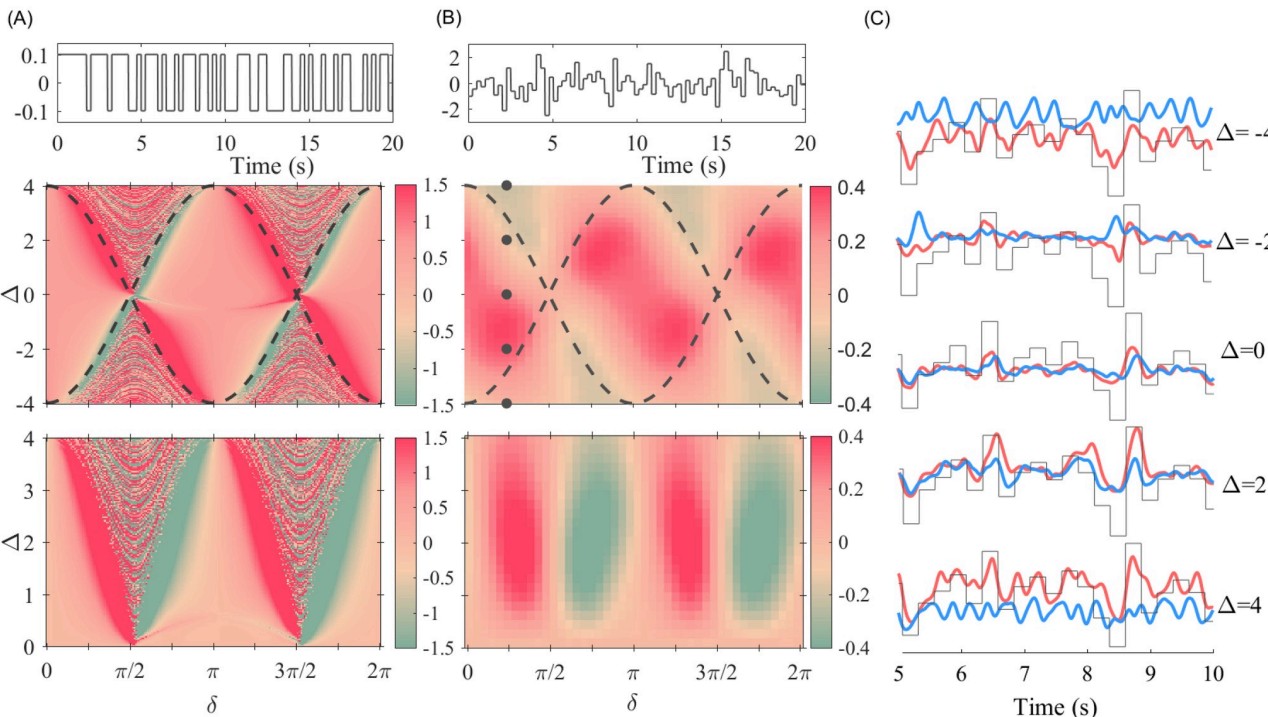

**Fig 8. Numerical results for the case of two Kuramoto oscillators.** In panel (A) top, we plot the stochastic dichotomous signal that is applied to the first oscillator (the sender). In panel (A) middle, we plot the correlation of the rate of the phase change of the second oscillator (the receiver) with the signal. The bottom panel is the transmission imbalance (Eq (13)) which quantifies the difference of the signal transmission in the two directions. In panel (B) top, we plot the random signal that is applied to the sender oscillator. In the middle panel, we plot the rate of the phase change of the receiver oscillator with the signal, and the bottom panel is the transmission imbalance. (C) Rate of phase change of two coupled oscillators for $\delta = \pi/4$ and $\Delta = 0$, $\pm 2$, $\pm 4$. The black line is the scaled input signal. The simulation parameters: $\omega = 55 Hz$, $K = 4$, $\delta \in [0, 2\pi]$, $\Delta = 4$, and we added noisy input with $\mu = 0$, and $\sigma = 1$.

high-frequency oscillator. Interestingly, the maximum response (and the minimum response in the reverse direction) is found near the boundaries of the locking zone.

To check the validity of our analytical results for the pulse packets and rate signals and the robustness of the results to the type of signal, we performed numerical simulations for two delay-coupled phase oscillators with $Q(\theta) = sin(\theta)$. We then applied a small amplitude dichotomous random signal, which switched between two states at random times, on the sender and calculated the correlation of the rate of the oscillations at the receiver with the input signal. The results shown in Fig 8A closely match the analytical ones (Fig 7C), within the locking region (whose borders are marked by dashed lines). An interesting point is that the numerical results, out of and near the locking region, reveal a reliable transmission and large imbalance between the transmissions in the two directions. However, since the analytical results were obtained assuming phase-locking, they are not valid for this region. Extension of the results to this region can be explained by the presence of intermittent locked epochs when the system approaches the locking zone [58].

Although the results normally depend on the exact form of PRC, a qualitative agreement with the numerical results for two neural populations (Fig 3) can be observed since the pPRC of the population is type-II. But since the two PRCs are not exactly the same, the results do not conform in details, e.g., the transmission is not symmetric in the two directions over different delay ranges and favor fast-to-slow direction (compare Fig 8 with Fig 3).

## Discussion

Extensive experimental and theoretical studies over recent decades have unleashed the role of brain oscillations in several cognitive and executive brain functions like sensory processing [59], memory [60, 61] attention [62], and motor functions [63]. Oscillations and the coordinated activity of neurons facilitate the transmission of signals along different stages of neural processing systems and enable an efficient communication between brain areas [64–66]. Integration of information which is processed across distributed specialized brain regions, is hypothesized to be controlled by the temporal coordination of their local dynamics [67]. Oscillations change the excitability of the neurons over time and enable control of communication between brain areas by adjusting their phase relationships [18]. In addition, oscillations provide a functional substrate to transmit multiple information along different routes and directions over different frequency bands [68–71]. Numerous theoretical and numerical studies have been carried out to shed light on the mechanisms through which the oscillations control the transmission of information and give rise to flexible communication channels in brain circuits [19, 20, 65, 72]. It was has been shown that the phase relationship between the oscillatory activity of brain regions can determine the efficacy and the preferred direction of the information flow [24, 26, 46, 52]. Diverse and dynamic phase relationships between the activity of different brain regions are a hallmark of a flexible pattern of communication in brain circuits [5]. In recent years, the mechanism that determines these phase relationships has been the focus of several theoretical studies addressing the control of information routing in brain circuits [19, 20, 26, 73, 74]. It is well-known that a mismatch between the natural frequencies exerts a finite phase difference between coupled oscillators operating in the locked state. This has been observed in coupled neural oscillators and for interconnected oscillatory populations of neurons [30, 33, 72]. In the case of zero, or very short, communication delay the oscillators with higher natural frequency tend to phase lead when the in-phase state is the stable mode [33]. These phase leaders have been usually considered as those that determine the preferred direction of information flow [19, 26]. In this framework, it is possible to tailor phase differences by changing the natural frequency of the neural populations. The oscillating frequencies can be modified by several networks and external parameters [11, 75]. Since the networks' parameters, like the synaptic strengths and time constants, are less (or are not even) tunable in short time scales, the external inputs to the networks are ideal candidates for controlling the oscillation frequency of the populations, at least in theoretical models [11, 75]. Therefore, the level of the external input can directly control the phase differences and the efficacy and direction of information transfer channels.

In this paper, we showed the essential role that the transmission delay plays in the efficiency and the direction of functional interaction between oscillating regions, when their natural frequencies are different. By systematically changing the natural-frequency mismatch and the delay time in the coupling between two oscillating neuronal populations, we showed that previous results on the information flow from high-frequency to low-frequency oscillating populations is only correct for small delays [20, 26]. Interestingly, other patterns of effective communication, including almost symmetric communication channel or information flow in the reverse direction, from the low- to the high-frequency population, can be observed over different ranges of delay. Our results indicate that phase-leading populations are not necessarily the source of information flow in brain networks. Parameters as the delay and the response function of the populations are important to determine this information flow.

It is well accepted that information can be processed and transmitted in brain circuits either using the rate coding or the time coding [50, 51]. To account for these two modes, we applied both slowly varying signals and synchronous pulse packets to the neural populations.

Interestingly, we found that our results were qualitatively the same for both types of signals. That is, the quality of the signal transmission as a function of the delay and frequency mismatch was similar in the two cases.

## Role of collective phase response functions

The PRC has been extensively shown to be important to explain different synchronization scenarios between coupled dynamical systems (see .e.g. [24, 30, 76, 77]). While the collective phase response is widely studied for populations of oscillators [78–81], those results are not readily applicable to the neural ensembles. The neural populations composed of excitatory and inhibitory neurons show collective oscillations at the population level while the single neurons irregularly fire [11, 82]. Since the mechanism of the synchrony and mathematical formulations of the emergence of the rhythms in such networks are far different from those of networks of coupled oscillators [75, 83]; though their response to the external inputs might be different and warrants for more systematic studies [24, 56, 84].

In the case of delay-coupled neuronal populations, the PRC can accurately determine the phase difference between populations operating in the locked state [24, 47]. A symmetric system composed of two identical populations in the presence of delayed interactions can exhibit in-phase or anti-phase-locking, or locking in other phases due to a symmetry breaking. Notably, a symmetry broken locking leads to a directional effective connectivity with the population that advances in phase playing the role of the information source. The role of symmetry breaking in the determination of the directional effective connectivity has been also reported in other studies [52]. These results confirm that phase-locking in a phase other than zero or $\pi$ can lead to a directional effective connectivity even if the underlying system is structurally symmetric and the populations have equal natural frequencies. The focus of our study was to provide a general framework for the quality and the preferred direction of the signal transmission between neural populations, in presence of delay and frequency mismatch. Our theoretical framework went beyond the determination of the phase differences, and showed how the transmission of rate- and time-coded signals non-trivially depend on the delay and frequency mismatch. We showed that once the collective PRCs of the neural populations are known, the regions to transmit signals efficiently can be predicted in the time delay vs. frequency detuning phase space. These results were in good agreement with those obtained when modulating the sender population and computing the information flow.

Finally, although the theoretical results were valid in the locked state, simulation results showed that they can be extrapolated to the region out of, and near to, the locked state (see Fig 8). We hypothesize that the presence of intermittent synchrony in the transition regions leads to the epochs of transient phase-locking where the theoretical results can be still valid. This can be of importance in realistic models of cortical activity where the synchronization and phase-locking modes are not stable and bursts of phase-locked strong oscillations appear between periods of low synchrony states [19].

## Role of the frequency detuning and connection delay

Based on the theory of communication through coherence (CTC) it is the phase difference between brain local oscillations that determines the efficacy of communication channels between brain regions [1, 20]. So, the study of the phase-locking between collective local oscillations in brain circuits has attracted much interest in recent years [24, 42]. Theoretical and numerical studies using networks of spiking neurons and low-dimensional mass models have shown that the delay in homogeneous systems [24, 42], and the frequency mismatch [33] affects the synchronization properties of brain networks. In general, in-phase and

anti-phase-locking are the main stable states of coupled neural oscillators in a symmetric system with delayed interaction (see [24, 46]), and in the presence of mismatch between the natural frequencies with the faster population leading the dynamics [42].

We found that such faster (phase leading) populations are not necessarily the source of information flow as was shown by several studies [19, 20, 26, 73, 74]. Depending on the delay, the preferred direction of signal transmission and information transfer can be established from fast-to-slow or in the reverse direction. Our results showed that, in general, it is the combination of the natural frequency detuning and the connection delay which matters and determines the effective connectivity.

Based on our results, for a symmetric interaction function like a sinusoidal one in Kuramoto model, there is a balance between two directions of effective connectivity in the parameter space. But as our numerical results showed, for two coupled neural populations the shape of the collective PRC warrants that there is a certain preference towards a larger flow of information for positive detuning than for negative ones, compatible with previous studies. Still, symmetric information exchange and information flow from the low to the high frequency population is also possible in some narrow range of delays.

### Limitations and future studies

In this study, our network operated at an oscillatory state with a single frequency. We showed that the signals and information transmission change with delay, when the latter is changed over a period of oscillation. At a given delay time, then, the transmission changes with the oscillation frequencies. It can be concluded that the effective connectivity between brain regions is different at different frequencies because of the delayed interactions. This provides the possibility to transmit information over different routes and directions at different frequency bands as is observed in experimental studies [68, 69]. Since our network was capable to produce oscillation in single frequency in the gamma range, it was not possible to check the transmission in multiple frequency bands.

In the brain, networks operate at multiple frequency bands (including a fast and a slow oscillatory component); several modeling studies have suggested mechanisms to reproduce this regime [85–87]. In our model we only considered synapses mediated by $AMPA$ and $GABA_A$ receptors, and used the classical description of the Hodgkin-Huxley neuron model. To account for networks with richer dynamics and to produce oscillations at multiple frequencies, we need to incorporate synapses with slow dynamics and chose more appropriate neuronal models [85, 88]. Moreover, to highlight the role of delay we set our network in a high coherence regime. It is necessary to explore how the results translate to more realistic networks with unstable oscillatory dynamics and lower coherence [19].

### Author Contributions

**Conceptualization:** Aref Pariz, Ingo Fischer, Alireza Valizadeh, Claudio Mirasso.

**Data curation:** Aref Pariz, Ingo Fischer, Alireza Valizadeh, Claudio Mirasso.

**Formal analysis:** Aref Pariz, Ingo Fischer, Alireza Valizadeh, Claudio Mirasso.

**Investigation:** Aref Pariz, Alireza Valizadeh, Claudio Mirasso.

**Methodology:** Aref Pariz, Alireza Valizadeh.

**Project administration:** Claudio Mirasso.

**Supervision:** Ingo Fischer, Alireza Valizadeh, Claudio Mirasso.

**Validation:** Alireza Valizadeh.

**Visualization:** Aref Pariz.

**Writing – original draft:** Aref Pariz, Ingo Fischer, Alireza Valizadeh, Claudio Mirasso.

**Writing – review & editing:** Ingo Fischer, Alireza Valizadeh, Claudio Mirasso.

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
