## [Decision Letter · Decision Letter 0]

5 Nov 2020

Dear Dr. Valizadeh,

Thank you very much for submitting your manuscript "Transmission delays and frequency detuning can regulate information flow between brain regions" for consideration at PLOS Computational Biology.

Let me apologize for the delay with which this decision is reaching you. These are difficult times, and a series of unfortunate coincidences made your paper end in the long tail of the distributions of editorial turnout times.

As with all papers reviewed by the journal, your manuscript was reviewed by members of the editorial board and by several independent reviewers. In light of the reviews (below this email), we would like to invite the resubmission of a significantly-revised version that takes into account the reviewers' comments.

Some of the concerns are major, and also point to a limited novelty and significance of the biological insight.

At this stage we prefer to give you the opportunity to revise it, but even if the technical issues are addressed, should the concerns about novelty and significance remain, we will propose conditional acceptance in PLOS One.

We cannot make any decision about publication until we have seen the revised manuscript and your response to the reviewers' comments. Your revised manuscript is also likely to be sent to reviewers for further evaluation.

Sincerely,

Daniele Marinazzo

Deputy Editor

PLOS Computational Biology

Daniele Marinazzo

Deputy Editor

PLOS Computational Biology

Reviewer's Responses to Questions

**Comments to the Authors:**

Reviewer #1: The review is uploaded as an attachment

Reviewer #2: The authors analyze the impact of transmission delays and frequency detuning for the information flow between brain regions. For this they use 2 populations of Hodgkin Huxley neurons and analyze the coherence, mutual information and the phase response curves. The manuscript is well written and seems technically sound and the results are sound and interesting, but a link to the literature on the impact of time-delays in oscillatory brain network models and neuronal fields is missing.

Below are my specific comments:

- Neural field dynamics with local and global connectivity and time delay has been studied for more than 10 years, see e.g. Jirsa PTRSA 2009.

- The impact of time-delays for the large scale brain dynamics has also been extensively studied, both analytically and computationally. Authors should make an effort to discuss their results also in the light of the findings about the impact of time-delays and the frequency missmatch on the synchronization of Kuramoto oscillators in brain networks (Petkoski et al PLOS CB 2018), which also hold for other oscillators (Petkoski et al. PTRSA 2019). Similarly, implicit time delays in KM (with phase shifts) were studied by Moon et al Plos CB 2015.

- how is the oscillation frequency in Fig 1 defined? Is it a simple ensemble average of frequencies of the firing for all the neurons, or is it the frequency of the locking? For phase oscillators these two can be different in asymmetric networks (as it is the case here with the E/I imbalance), see e.g. Petkoski et al PRE 2013.

- would the results be different if there are no time delays within the populations? This approach is for example common when using brain network model, e.g. Sanz Leon Neoroimage 2015.

- Can the authors elaborate on the advantage of using populations of neurons instead of neuronal masses such as Wong Wang or Jansen Ritt models? Do they expect at least the PRC and coherence results to be fully confirmed if neuronal masses are used?

- the way delta is defined in Eq. 6, requires all the oscillators to be synchronized. Isn’t this too strict condition, and wouldn’t it be more general if the time-delays are explicit in the coupling function, so that the phase shift as defined by the authors still holds, but only for the locked neurons? Similarly, for the simulation of the Kuramoto oscillators, are the the time-delays explicit in the interactions, or there are phase-shifts instead, implying the model to be valid only if all the neurons are synchronized?

- can it be stated more clearly what is the impact of the assymetry in the coupling, as it was done for example in the above mentioned similar works by Moon et al Plos CB 2015 &Petkoski et al Plos CB 2018?

- line 395 typo, “In” instead of “If”.

Taken together, I cannot recommend publication of the manuscript in the current form.

**Have all data underlying the figures and results presented in the manuscript been provided?**

Reviewer #1: **No: **There is any extra data. But I think it is not necessary.

Reviewer #2: None

PLOS authors have the option to publish the peer review history of their article (what does this mean?). If published, this will include your full peer review and any attached files.

Reviewer #1: No

Reviewer #2: No
---

## [Decision Letter · Decision Letter 1]

16 Feb 2021

Dear Dr. Valizadeh,

We are pleased to inform you that your manuscript 'Transmission delays and frequency detuning can regulate information flow between brain regions' has been provisionally accepted for publication in PLOS Computational Biology.

Best regards,

Daniele Marinazzo

Deputy Editor

PLOS Computational Biology

Reviewer's Responses to Questions

**Comments to the Authors:**

Reviewer #1: Review is uploaded as an attachment

Reviewer #2: The authors have addressed most of the issues I have raised and I recommend the article to be published in the current form.

**Have all data underlying the figures and results presented in the manuscript been provided?**

Reviewer #1: None

Reviewer #2: None

PLOS authors have the option to publish the peer review history of their article (what does this mean?). If published, this will include your full peer review and any attached files.

Reviewer #1: No

Reviewer #2: No

---

## [Editor Report · Acceptance letter]

25 Mar 2021

PCOMPBIOL-D-20-01170R1 

Transmission delays and frequency detuning can regulate information flow between brain regions

Dear Dr Valizadeh,

I am pleased to inform you that your manuscript has been formally accepted for publication in PLOS Computational Biology. Your manuscript is now with our production department and you will be notified of the publication date in due course.

With kind regards,

Andrea Szabo
